# Anti-Inflammatory Effects of Pomegranate Peel Extracts on In Vitro Human Intestinal Caco-2 Cells and Ex Vivo Porcine Colonic Tissue Explants

**DOI:** 10.3390/nu11030548

**Published:** 2019-03-05

**Authors:** Fabio Mastrogiovanni, Anindya Mukhopadhya, Nicola Lacetera, Marion T. Ryan, Annalisa Romani, Roberta Bernini, Torres Sweeney

**Affiliations:** 1Department of Agriculture and Forest Sciences (DAFNE), University of Tuscia, 01100 Viterbo, Italy; fabio.m@unitus.it (F.M.); nicgio@unitus.it (N.L.); 2School of Veterinary Medicine, University College Dublin 4, D04 V1W8 Dublin, Ireland; anindya131283@gmail.com (A.M.); marion.ryan@ucd.ie (M.T.R.); torres.sweeney@ucd.ie (T.S.); 3Department of Statistics, Computing, Applications “G. Parenti” (DISIA), PHYTOLAB, University of Florence, 50019 Florence, Italy; annalisa.romani@unifi.it

**Keywords:** anti-inflammatory effects, human colonic adenocarcinoma cells (Caco-2 cells), porcine colonic tissues, pomegranate (*Punica granatum* L.) peel extracts, polyphenols, punicalagin, circular economy

## Abstract

The aim of this study was to determine the anti-inflammatory potential of pomegranate peel extracts (PPE) prepared from waste material of pomegranate juice production both in vitro on Caco-2 cells and ex vivo using porcine colonic tissue explants. Caco-2 cells were stimulated in vitro by TNF and colonic tissue explants were stimulated ex vivo with lipopolysaccharide (LPS). Both tissues were co-treated with PPE at 0, 1.0, 2.5, 5.0, 10 and 25 μg/mL. The secretion of CXCL8 in the supernatant of both experiments was determined by enzyme linked immunosorbent assay (ELISA) and the relative expression of inflammatory cytokines were evaluated in the colonic tissue by quantitative polymerase chain reaction (QPCR). The 2.5 to 25 μg/mL of PPE suppressed CXCL8 (*p* < 0.001) in the Caco-2 cells, whereas CXCL8 production was suppressed by only 5 and 25 μg/mL (*p* < 0.01) of PPE in the colonic explants. The 5 μg/mL of PPE also suppressed the expression of *IL1A* (*p* < 0.05), *IL6* (*p* < 0.01) and *CXCL8* (*p* < 0.05) in LPS challenged colonic tissues compared to controls. In conclusion, the 5 μg/mL of PPE consistently elicits strong anti-inflammatory activity. These results support the potential of bioactive compounds from the waste peel of pomegranate in terms of their anti-inflammatory activity in cells and tissues of the gastrointestinal tract.

## 1. Introduction

Epidemiological studies have identified an inverse relationship between the incidence of a range of common non-communicable diseases and the consumption of fruits and vegetables that contain valuable bioactive compounds; a relationship that is consistent with the principles of the Mediterranean diet [1,2,3,4]. Foods rich in polyphenols have attracted interest due to their beneficial effects on human health, with evidence of antioxidant [5,6], anti-cancer [7,8,9,10], cardio-protective and anti-inflammatory activities [11,12,13]. Fruits are used commonly for juice or pulp extraction, a process which generates large amounts of waste containing valuable bioactive compounds [14,15]. The recovery of these bioactive molecules and their reuse within agricultural, food, cosmetic and pharmaceutical industries supports the core principles of the “circular economy” [16]. This recently conceived economic model is based on sustainable processes at “zero waste” that we have already described in previous studies [17,18,19].

*Punica granatum* L., commonly known as pomegranate, is consumed worldwide as either fresh fruit or in the form of juice and is recognised for its antioxidant, antimicrobial, anti-inflammatory and anti-cancer health benefits [20,21,22,23]. Originally native to Iran, over time the fruit has gained increased popularity and are now produced in a number of countries including Tunisia, Turkey, Spain, Italy, Egypt, Morocco, United States, China, India, Argentina, Israel and the South Africa. The variety “Wonderful” is the most widely cultivated due to its high yield and quality [24]. The fruit consists of three main parts: the arils, the seeds and the peel, each contributing 40%, 10% and 50% of the total fruit weight, respectively. The arils contain discrete quantities of flavonoids, principally anthocyanins, which are responsible for the red colour of the juice. The seeds and seed oil contain high levels of polyunsaturated fatty acids. The peel, the main waste component of pomegranate juice production, is particularly rich of hydrolysable tannins, specifically punicalagins, which are multiple esters of gallic acid and glucose, widely known for their antioxidant and anti-inflammatory properties [25,26,27,28,29,30,31]. 

Several disorders related to the chronic inflammation of the gastrointestinal tract (GIT), including ulcerative colitis and Crohn’s disease are classified broadly as Inflammatory Bowel Disease (IBD) [32]. A range of genetic, immunological and environmental factors are thought to be responsible for the development of IBD and its incidence is increasing throughout the world [33]. Numerous studies have illustrated the imbalance of cytokines during acute and chronic inflammation and their role in the modulation of the immune response and resolution of inflammation. Un-regulated immune activation in the GIT drives chronic inflammation and disease [34]. Hence, immunomodulatory supplements that support GIT health are constantly being sought and polyphenols found in plant-based foods could be effective in this regard [35,36,37].

In this study, the anti-inflammatory effects of standardised pomegranate peel extracts (PPE) were investigated using an in vitro model where human colonic adenocarcinoma Caco-2 cells were stimulated with TNF and an ex vivo model using porcine colonic tissues stimulated with LPS. The effects were studied using non-cytotoxic concentrations of PPE on Caco-2 cell lines (1.0, 2.5, 5.0, 10 and 25 μg/mL). Caco-2 cells represent the epithelial layer of the small intestine, they have a functional resemblance to the colonic enterocytes and an ability to respond to pro-inflammatory stimuli; hence this in vitro system is a useful model of the human intestine with regard to screening potential bioactive compounds [37,38]. One caveat however, is that this in vitro model lacks the cellular heterogeneity present in the colon in vivo [39]; hence, to circumvent this limitation, ex vivo intestinal tissues explants were used as a validation model as they are informative with regard to physiological and immunological responses both under normal and challenged conditions [39,40]. The total RNA in porcine colonic explants remains intact for up to 3 h post-mortem, hence this ex vivo system can be used to explore and validate the effects of bioactive compounds on the transcriptome in the presence or absence of inflammatory stimulants [41]. 

The objective of this current study was to evaluate the anti-inflammatory effects of incremental PPE concentrations (1.0, 2.5, 5.0, 10 and 25 μg/mL) on an in vitro model using Caco-2 cells stimulated with TNF and ex vivo model using porcine colonic tissues stimulated with LPS. To the best of our knowledge, this paper describes for the first time the anti-inflammatory effects of PPE in an ex vivo porcine colonic model.

## 2. Materials and Methods

### 2.1. Plant Material, Extraction Procedure and Chemical Characterisation

*Punica granatum* L. (“Wonderful” variety) fruits were collected in Grosseto (Tuscany, Italy; Latitude: 42°45′46” N; Longitude: 11°06′33” E). The pomegranate peel extract (PPE) was prepared as already described [42,43,44]. Briefly, the crushed peel was treated with ultrapure water (weight/volume = 1/12) at 100 °C, the residue was filtered and water removed by lyophilisation using LYOVAC GT 2 (Leybold AJ-Cologne, Germany).

High Performance Liquid Chromatography (HPLC)/Diode Array Detection (DAD)/Electron Spray Ionisation (ESI)-Mass Spectrometry (MS) analyses were carried out using a HP-1100 liquid chromatograph equipped with a DAD detector, a HP 1100 MSD API-electrospray (Agilent Technologies-Palo Alto, CA, USA) detector, a Luna C18 column 250 × 4.60 mm, 5 µm (Phenomenex-Torrance, CA, USA) using the analytical conditions already described [44]. The Nuclear Magnetic Resonance (NMR) spectra were recorded using a 400 MHz nuclear magnetic resonance spectrometer Advance III (Bruker, Germany) [42,43]. 

### 2.2. Caco-2 Cell Culture

Caco-2 cells were supplied by the American Type Culture Collection (ATCC) and maintained in 75 cm^2^ cell culture flasks and Dulbecco’s Modified Eagle’s Medium (DMEM) (Invitrogen, San Diego, CA, USA) supplemented with 10% (*v*/*v*) fetal bovine serum (Invitrogen Corp.), 1.0% sodium pyruvate, 1.0% non-essential amino acids and 1.0% penicillin-streptomycin (Sigma-Aldrich, St. Louis, MO, USA) at 37 °C in a humidified 5% CO_2_ incubator. 

#### 2.2.1. Cytotoxicity Evaluation

To evaluate the cytotoxic activity of PPE, the cells were plated into 96-well microplates at 1 × 10^6^ cells/mL and maintained for 21 days until fully differentiated. After that, the media was replaced with different concentrations of PPE (1.0, 2.5, 5.0, 10, 25 µg/mL) and one aliquot of cells was treated with regular medium (control). The cell culture plates were incubated at 37 °C for 24 h. Cell viability was measured at the end of the exposure time, using an MTT (3-(4,5-dimethylthiazol-2-yl)-2,5-diphenyl tetrazolium bromide) assay, in the following way: all treatments were replaced with 100 μL of MTT solution (regular medium and MTT) to 0.5 mg/mL of final concentration and the plates were kept again in incubator for 3.5 h; after incubation, the supernatants were discarded and 100 μL of dimethyl sulfoxide (DMSO) were added to each well; plates were additionally incubated for 45 min and subsequently read at 595 nm by microplates reader, absorbance data were recorded.

#### 2.2.2. In Vitro Interleukin 8 (CXCL8) Detection using Enzyme Linked Immunosorbent Assay (ELISA)

For CXCL8 detection from Caco-2 cells, the cells were plated in a 24 well plate at 10^6^ cells mL^−1^ of density and maintained for 21 days until fully differentiated. The media was changed on alternative days. The cells used in this study were between the 53rd to 65th passages. After 21 days, an aliquot of cells was treated with DMEM and 10 nM Dexamethasone (Dexa) (Sigma-Aldrich) a commercially available anti-inflammatory agent; another aliquot was treated with only DMEM (Control); the other cells were treated with DMEM containing increasing concentrations of PPE (1.0, 2.5, 5.0, 10, 25 µg/mL). All treatments contained 10 nM TNF (Sigma-Aldrich) with the aim of inducing a pro-inflammatory response. After 24 h of treatment, supernatants were removed and CXCL8 concentrations were measured using the Human CXCL8 ELISA kit (R&D Systems Europe, Ltd. Abingdon, UK). 

### 2.3. Ex Vivo Porcine Colonic Tissue Collection and Treatment

Six 38 day-old pigs were sacrificed and colonic tissues were dissected and washed using phosphate buffer saline (PBS). From each colonic tissue, sections of about 1.0 cm^2^ of tissue were taken and the smooth muscle was stripped from the overlying epithelium and discarded. For each animal, one section was immediately placed in 15 mL of RNAlater^®^ (Applied Biosystems, Foster City, CA, USA) as time-zero control (T_0_) and stored at room temperature. Another tissue section was incubated in DMEM (Control) and further tissue sections were incubated in different wells, each one containing DMEM with increasing concentrations of PPE (1.0, 2.5, 5.0, 10, and 25 µg/mL). To induce an inflammatory response, each treatment, with the only exception of T_0_, contained 10 µg/mL of LPS (Sigma Aldrich). All tissue explants were incubated at 37 °C for 3 h and following this, the supernatants were stored at −20 °C, surplus media was removed from the tissue and the tissue was stored in 15 mL of RNAlater^®^ overnight at room temperature. The RNAlater^®^ was then removed and the samples stored at −80 °C.

#### 2.3.1. Ex Vivo Interleukin 8 (CXCL8) Detection using ELISA

The CXCL8 concentrations were measured in the supernatants collected 3 h post LPS stimulation of the colonic explants using the Porcine CXCL8 ELISA kit (R&D Systems Europe, Ltd. Abingdon, UK).

#### 2.3.2. RNA Extraction and cDNA Synthesis 

Total RNA was extracted from colonic tissues using Trizol (Sigma-Aldrich) with a further purification using GenElute™ Mammalian Total RNA Miniprep Kit (Sigma-Aldrich), according to the manufacturer’s instructions. During the process, genomic DNA was removed using DNAse I (Sigma-Aldrich). Total RNA was quantified using a NanoDrop-ND1000 Spectrophotometer (Thermo Fisher Scientific Inc., Waltham, MA, USA). The cDNA was synthesised using First Strand cDNA Synthesis Kit (Qiagen Ltd. Crawley, UK), 1.0 μg of total RNA and oligo dT primers, in accordance with the manufacturer’s instructions. The final volume of cDNA was adjusted to 250 μL with nuclease free water.

#### 2.3.3. Quantitative Real-Time PCR (QPCR) 

The relative expression of a panel of nine cytokines were evaluated using QPCR: interleukin 1 alpha (*IL1A*), interleukin 1 beta (*IL1B*), interleukin 6 (*IL6*), interleukin 8 (*CXCL8*), interleukin 10 (*IL10*), interleukin *17A* (*IL17A*), interferon gamma (*IFNG*), Tumor Necrosis Factor alpha (*TNF*) and Transforming Growth Factor beta 1 (*TGFB1*). Two endogenous controls, β2 microglobulin (*B2M*) and Beta-actin (*ACTB*) were used to normalize data [39]. Primers used for these targets are described in Table 1. Primer efficiencies were determined using a serial dilution (1:4 dilution series over 7 points) of a cDNA pool from all the experimental samples and were in the range of 90 to 110%. All primers were designed using Primer Express™ software and were synthesised by MWG Biotech (Milton Keynes, UK). Assays were carried out using 96 well fast optical plates on a 7500HT ABI Prism Sequence Detection System (PE Applied Biosystems, Foster City, CA, USA) using Fast SYBR Green PCR Master Mix (Applied Biosystems). All reactions were performed in duplicate to a final volume of 20 μL containing 10 μL Fast SYBR PCR Master mix, 1 μL forward and reverse primer mix (100 μM), 8 μL nuclease free water and 1 μL of cDNA. The thermal cycling conditions were 95 °C for 10 min followed by 40 cycles of 95 °C for 15 s and 65 °C for 1 min followed by a dissociation curve analysis to confirm specificity. 

The stability of the reference genes and normalisation factors (NF) were calculated using the qbase^+^ package (Biogazelle, Gent, Belgium) based on the geNorm algorithm. Briefly, geNorm calculated the stability measure M for a reference gene as the average pairwise variation (V) for that gene with all other tested reference genes. A Vn/n+1 value was calculated for every comparison between two consecutive numbers (n and n+1) of candidate reference genes. Following the stepwise exclusion of the least stable reference genes, M values were re-calculated and the stability series obtained. Finally, the normalisation factor was calculated from the two most stable genes (*B2M*, *ACTB*), as the geometric mean of the most stable reference genes, and the normalised relative quantity (RQ) of the target genes is obtained which is an expression of the ratio between the RQ and the sample specific NF. The basic formula for relative quantification (RQ = 2^ddCt) assumes 100% amplification efficiency (E = 2). 

### 2.4. Statistical Analysis

All data were normalised and analysed as a complete randomised design experiment using one way ANOVA function of GraphPad PRISM software. Spearman’s rank correlation analysis was performed to correlate the expression of the selected panel of cytokine expression post LPS challenge. Probability values of *p* <0.05 were used as the criterion for statistical significance. All results are reported as square means ± standard error of the means.

## 3. Results

### 3.1. Polyphenolic Composition of PPE

The polyphenol composition of PPE is presented in Figure 1. Both high-molecular and low-molecular weight phenolic compounds are present. High-molecular weight phenols were α- and β-punicalagin with 146.9 ± 1.465 mg/g and 266.3 ± 1.687 mg/g, respectively; gallic acid, ellagic acid and granatin B were low-molecular weight phenols found into PPE as minor components, as published previously [42,43]. 

### 3.2. Experimental Results on Caco-2 cells

#### 3.2.1. Cytotoxicity on Caco-2 cells

The cytotoxic effects of PPE on Caco-2 cells are presented in Figure 2. No cytotoxic effects were observed in Caco-2 cells treated for 24 h with PPE at any concentrations tested 1.0, 2.5, 5.0, 10 and 25 μg/mL (*p* > 0.05).

#### 3.2.2. CXCL8 Concentrations in Caco-2 cell Supernatants

The effects of the treatment of Caco-2 cells with TNF alone (positive control) or in combination with either Dexamethasone (negative control) or increasing concentrations of PPE (1.0 µg/mL to 25 µg/mL) on CXCL8 secretion over 24 h is presented in Figure 3A. TNF challenged Caco-2 cells treated with Dexamethasone had a 57% reduction in CXCL8 production compared to the positive control (*p* < 0.001). The TNF challenged Caco-2 cells treated with 2.5, 5.0, 10 and 25 μg/mL of PPE compared to the positive control cells had a reduction of 28% (*p* < 0.001), 35% (*p* < 0.001), 43% (*p* < 0.001) and 40% (*p* < 0.001), respectively. There were no differences observed with treatment with 1.0 μg/mL PPE (*p* > 0.05).

### 3.3. Experimental Results on Ex-Vivo Porcine Colonic Tissue Explants

#### 3.3.1. CXCL8 Concentrations in the Supernatant

The anti-inflammatory effects of PPE (1.0 µg/mL to 25 µg/mL) on freshly excised ex vivo colonic tissues co-stimulated with LPS are presented in Figure 3B. The CXCL8 concentrations in the supernatants decreased with 5.0 and 25 μg/mL of PPE, with a 59.3% (*p* < 0.01) and 54.5% (*p* < 0.01) reduction compared to the control, respectively. There were no differences observed with the other PPE concentrations tested (*p* > 0.05).

#### 3.3.2. Effect of LPS on Cytokine Expression in Porcine Colonic Tissues

The effect of LPS challenge on the expression of a selected panel of cytokine genes from the ex vivo colonic tissues are summarized in Figure 4. The expression of *IL17A* and *IL10* were not affected by the LPS challenge. Compared to the untreated control, the LPS challenge increased the expression of pro-inflammatory cytokines *IL1A*, *IL1B*, *IL6*, *CXCL8*, *IFNG* and *TNF* by 95.1% (*p* < 0.001), 84.7% (*p* < 0.001), 88.3% (*p* < 0.001), 88.3% (*p* < 0.001) and 84.2% (*p* < 0.01), respectively. In contrast, the expression of *TGFB1* was down-regulated (*p* < 0.01) compared to the untreated control.

#### 3.3.3. Effect of PPE on Cytokine Expression in LPS Challenged Porcine Colonic Tissues

The effect of increasing concentrations of PPE on LPS challenged porcine colonic tissue explants is presented in Figure 5. There were no differences observed in the expression of *IL10*, *IL17A* and *IFNG* between control and PPE treated tissues (Figure 5B). The co-treatment of LPS challenged porcine tissues with 5.0 μg/mL of PPE was associated with the highest reduction of *IL1A* by 54.3% (*p* < 0.05), *IL6* by 70.6% (*p* < 0.01) and *CXCL8* by 62.2% (*p* < 0.05) compared to the challenged control (Figure 5A). Similarly, co-treatment of LPS challenged tissue with 2.5 μg/mL of PPE was associated with a 67.9% (*p* < 0.05) reduction of *IL6* (Figure 5A).

A heat map representing the correlations (R^2^) of a selected panel of cytokines in ex vivo porcine colonic tissues post LPS challenge and co-treatment with PPE is presented in Figure 6. The expression of cytokine *IL1A* is positively correlated to *IL1B* and *TNF* (*p* < 0.05). Similarly, the expression of *CXCL8* is positively correlated to *IL6* expression (*p* < 0.001).

## 4. Discussion

The “circular economy” is an economic strategy that encourages the reuse and valorisation of waste using environmentally friendly workflows [16]. Hence the extraction of bioactive compounds from the waste products of the pomegranate juice industry, using water as a solvent, respects the main principles of “The Green Extraction of Natural Products” [45,46]. In this study, the anti-inflammatory properties of PPE were evaluated both in an in vitro Caco-2 cell model and in an ex vivo porcine colonic explant model. Of significance was the fact that none of the tested concentrations of PPE exhibited any cytotoxicity effects. Further investigations indicated that the higher concentrations of PPE exhibited strong anti-inflammatory effects in both in vivo and ex vivo models by suppressing CXCL8 concentrations in cell and tissue supernatants, respectively. The 5.0 µg/mL concentration of PPE provided the most consistent results across all measures as it was also associated with the highest suppression of pro-inflammatory cytokine (*IL1A*, *IL6* and *CXCL8*) expression in the ex vivo model. 

These data highlight the anti-inflammatory properties of pomegranate peel and support further in vivo studies on inflammatory conditions of the gastrointestinal tract and other organs of the body. Indeed, numerous therapeutic properties of pomegranate extracts have been characterised in a variety of in vitro, animal and clinical trials [47]. Recent studies have investigated the possible role of micronutrients and bioactive compounds found in plant-derived foods such as vitamins, and polyphenols in the management of IBD [35,36]. In clinical trials, curcumin and beverages rich in polyphenols as green tea have effectively reduced the symptoms in IBD patients [35]. To the best of our knowledge, punicalagins have not been included so far in these studies even if they are recognized as the most therapeutically beneficial pomegranate constituents [47]. 

Numerous studies have demonstrated that during both chronic unresolved and acute inflammation of the gastrointestinal tract, there is an imbalance in the inflammatory cytokine profile [33]. The inflamed intestinal immune system has elevated levels of pro-inflammatory cytokines, while the resolving anti-inflammatory cytokines are under-represented, often resulting in chronic inflammation and diseases such as IBD [32,34]. Both acute and chronic gut inflammatory conditions are characterised by up-regulation of pro-inflammatory cytokines, especially *TNF* and *CXCL8*. While TNF is considered to be one of the main pro-inflammatory mediators [48,49], CXCL8 (produced by immune cells macrophages, endothelial cells and epithelial cells) possess chemotactic activities targeting T lymphocytes, basophils and neutrophils [50]. Interestingly, in the current study, when Caco-2 cells were stimulated with TNF and co-treated with an increasing range of concentrations of PPE, a reduction in secretion of CXCL8 was observed. 

Our current findings are in accordance to a previous study by Hollebeeck et al. [51] that observed the anti-inflammatory properties of a polyphenol component of pomegranate husk, punicalagin, in Caco-2 cells. While this study also used Caco-2 cells, they induced inflammation using a cocktail of IL1B, TNF, IFNG and LPS after treating the cells with the pomegranate husk extracts for 1 h. Interestingly, the observations are similar in both studies, a reduction in pro-inflammatory cytokines (mRNA and proteins) are recorded, alluding the potential of PPE as anti-inflammatory agent. 

While the single layer of Caco-2 cells is suited for initial screening of bioactive compounds, it is also a limiting factor in understanding the mechanisms involved in vivo [40]. The ex vivo colonic models, reflecting the in vivo complexity of colonic tissue provides a greater insight compared to the Caco-2 cell in vitro model [52]. Several previous studies have been based on the concept that pigs are good models for human research and hence have used ex vivo porcine colonic explants to investigate the anti-inflammatory effects of seaweed extracts [38] and milk hydrolysates [40,53]. Furthermore, LPS, a bacterial endotoxin, is extensively used to mimic a microbial challenge. Bacterial LPS stimulates the TLR4/MD-2 complex and exerts its effects by stimulating pro-inflammatory cytokines in the tissue explants [54]. 

In this current study, LPS treatment of the porcine colonic tissues resulted in a significant increase in the expression of a panel of pro-inflammatory cytokines indicating a successful pro-inflammatory response against the LPS stimulus. Further to this, the anti-inflammatory properties of PPE in response to the external LPS stimuli was validated in this ex vivo system. While the stimulation of porcine colonic explants with LPS resulted in an up-regulation of both *IL1A* and *IL1B*, the co-treatment with 5.0 μg/mL concentration of PPE was associated with a reduction in *IL1A* expression only. The pro-inflammatory cytokine IL1A is produced by various cell types including monocytes/macrophages, neutrophils, and endothelial cells, which mediate immune and inflammatory responses and is associated to formation of inflammatory lesions in IBD patients [34]. This is in accordance with previous studies where suppression of *IL1A* expression was observed in LPS stimulated porcine colonic tissue co-treated with milk hydrolysate [39,53]. 

The pro-inflammatory cytokine IL6 plays a major role in inflammation, infection and cancer [55]. In healthy humans, the normal physiological concentrations of IL6 in serum are low; therefore, elevated IL6 concentrations in the serum indicate systemic inflammation or infection. Elevated IL6 in both the serum and mucosal biopsies is an indicator of IBD [56]. In this current study, the LPS challenge increased *IL6* expression in colonic explants. Interestingly, co-treatment of LPS challenged colonic explants simultaneously with 2.5 μg/mL and 5.0 μg/mL of PPE resulted in suppression of *IL6* expression. 

Similar to *IL6*, elevated expression of the pro-inflammatory cytokine *CXCL8* is observed in mucosal diseases [34]. CXCL8 is a powerful neutrophil chemoattractant and activator, produced by macrophages, epithelial cells, and fibroblasts in the intestinal mucosa [57]. The strong association observed in several studies between concentrations of CXCL8 and intestinal inflammation has highlighted this cytokine as an important clinical indicator of inflammation [58]. The decrease in CXCL8 concentrations observed in this study in response to PPE, both in supernatant of tissues stimulated with LPS and in the LPS challenged colonic explants, highlight its potential as an anti-inflammatory agent capable of lowering CXCL8 concentrations and even resolving inflammation in vivo. In addition, this study confirms the anti-inflammatory effects evidenced by the reduction of both CXCL8 protein and suppression of *CXCL8* gene expression. 

The suppression of *IL6* and *CXCL8* expression by PPE indicates that its anti-inflammatory potential is most likely mediated through the suppression of the nuclear factor kappa B (NF-κβ) pathway. In fact, a correlation analysis indicated that the expression of pro-inflammatory cytokines *IL1A*, *IL1B*, *IL6*, *CXCL8* and *TNF* were positively correlated with each other. The NF-κβ pathway plays a central role in the inflammatory response, specifically by regulating the expression of cytokines involved in the inflammation [59,60]. While polyphenols are known to suppress inflammatory responses by supressing NF-κβ and mitogen-activated protein kinase (MAPK) signalling pathways [61], the underlying mechanism of action of these compounds are still not well understood. In our previous study, using a bovine mammary cell model [43], we observed anti-inflammatory activity by the same PPE used in the current study. We suggested that the observed anti-inflammatory activity of PPE could be related to a bioactive hydrolysable tannin, the punicalagin, that was present in high amounts (88.8%) in our extract in form of α and β isomers [43]. Additionally, in a previous study [62], punicalagin acted on NF-κβ pathway as a suppressor of the phosphorylation of IκBα and p65, thus inhibiting the activation of LPS induced NF-κβ. Hence, we propose that the anti-inflammatory activity of PPE observed in this study in both in vitro and ex vivo models is highly associated to punicalagins present in PPE. 

This study also gives an insight into the use of an increasing concentrations of PPE in both in vitro and ex vivo models to optimise the best concentrations that can be used in further experiments. While none of the concentrations used in this study elicits any cytotoxic effect in the in vitro model, in the ex vivo model we observed a dose response, especially in the expression of cytokines *IL1A*, *IL6* and *CXCL8*. The 5.0 μg/mL concentration of PPE was associated with the highest reduction in expression of these pro-inflammatory cytokines whereas with concentrations above 5.0 μg/mL this anti-inflammatory effect is lost. A similar loss of bioactivity was observed in a previous study using pistachio polyphenol extracts on LPS stimulated monocyte/macrophage cell-line J774-A1; while 0.1 mg/mL supressed IĸB-α secretion, 0.5 mg/mL increased IĸB-α secretion [63].

Therefore, dose response studies such as the current study are of high importance to evaluate optimal doses to obtain bioactivities of potential food ingredients.

## 5. Conclusions

This study highlights the anti-inflammatory potential of a water extract of pomegranate peel (PPE). PPE reduced the secretion of the pro-inflammatory cytokine CXCL8 in TNF stimulated Caco-2 cells. In addition, PPE reduced CXCL8 secretion as well as suppressing the gene expression of pro-inflammatory cytokines (*IL1A*, *IL6* and *CXCL8*) from LPS challenged colonic tissues. These results suggest that PPE has the potential to be used as a functional food ingredient with applications in the maintenance of gastrointestinal tract homeostasis. While the data presented in this study have been generated in in vitro and ex vivo environments, they highlight the potential value of further studies exploring the prevention of inflammatory conditions of the gastrointestinal tract, such as IBD, with pomegranate extracts rich in punicalagins. 

## Figures and Tables

**Figure 1 nutrients-11-00548-f001:**
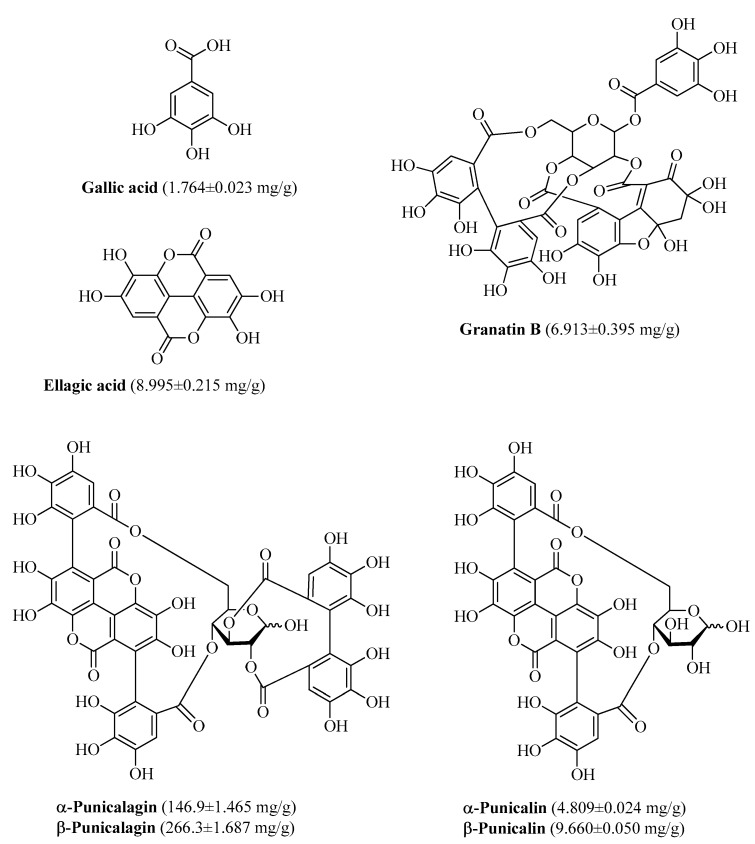
Phenolic compounds found in Pomegranate Peel Extracts (PPE) using High Performance Liquid Chromatography (HPLC) and Nuclear Magnetic Resonance (NMR) analyses. The HPLC analysis was carried out using a HP-1100 liquid chromatograph equipped with a Diode Array Detection (DAD) detector, a HP 1100 MSD API-electrospray (Agilent Technologies) detector and a Luna C18 column 250 × 4.60 mm. The NMR spectra were recorded using a 400 MHz nuclear magnetic resonance spectrometer Advance III.

**Figure 2 nutrients-11-00548-f002:**
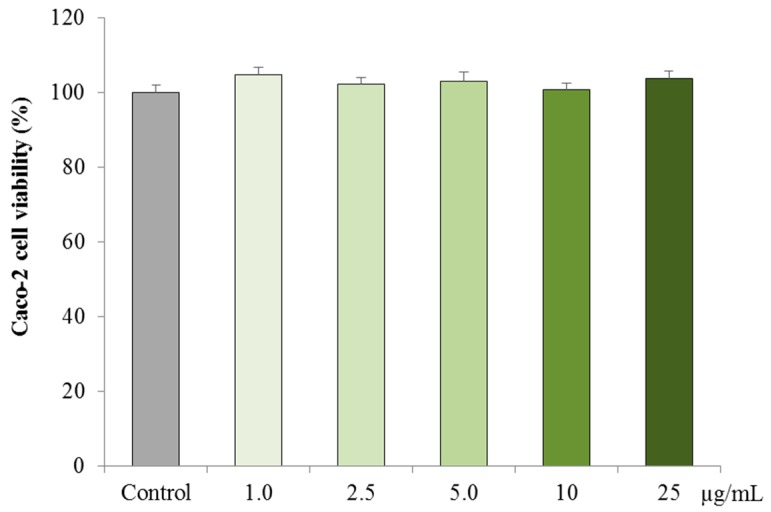
Effect of exposure to PPE on Caco-2 cell viability, post 24 h. Each data point represents *n* = 3; data presented as mean ± standard error. No significant differences were observed with PPE exposure compared to the control (*p* > 0.05).

**Figure 3 nutrients-11-00548-f003:**
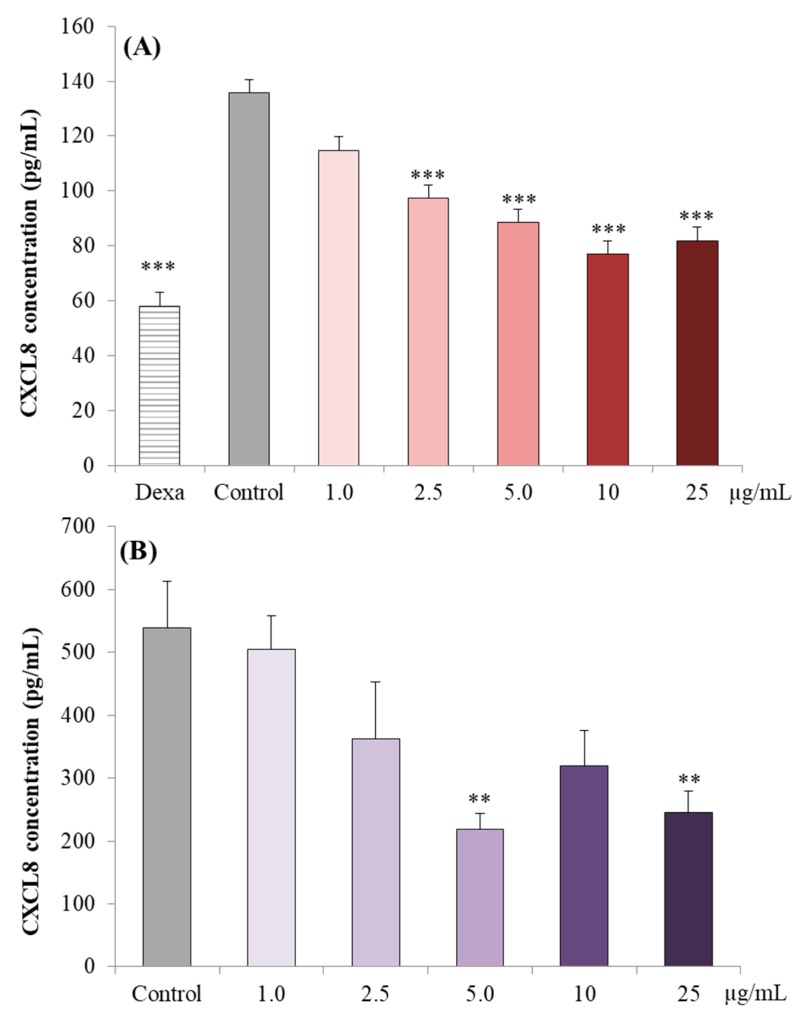
(**A**) Effect of PPE concentrations on CXCL8 concentrations in the supernatant of Caco-2 cells stimulated by TNF and (**B**) in the supernatant of porcine colonic tissue stimulated by lipopolysaccharide (LPS). Supernatants were collected from Caco-2 cells, 24 h post-incubation and from porcine colonic tissues, 3 h post-incubation. Each data point represents *n* = 3, data presented as mean ± standard error; Dexa = Dexamethasone; *** *p* < 0.05 and ** *p* < 0.01 in comparison to control treatment group.

**Figure 4 nutrients-11-00548-f004:**
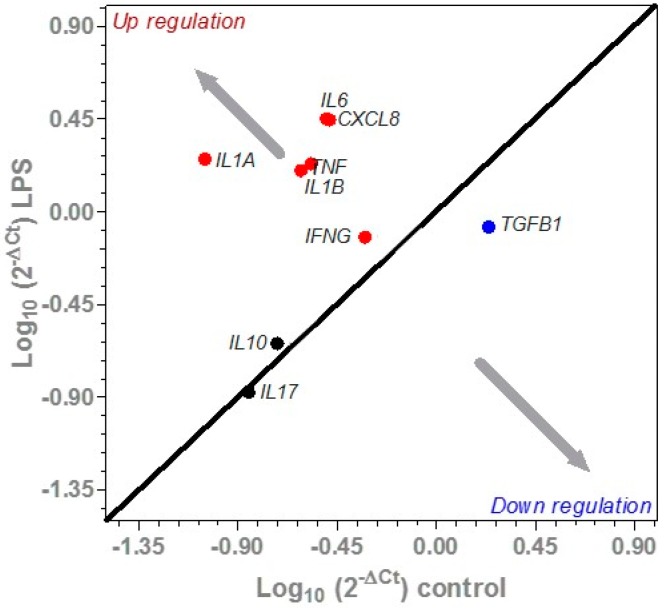
Differences in the expression of a selected panel of cytokine genes between unchallenged and lipopolysaccharide (LPS) challenged ex vivo colonic tissues. The red dots represent significant upregulated genes (*p* < 0.05), the blue dot represents a significant downregulated gene (*p* < 0.05) and the black dots represent genes not affected by LPS (*p* > 0.05).

**Figure 5 nutrients-11-00548-f005:**
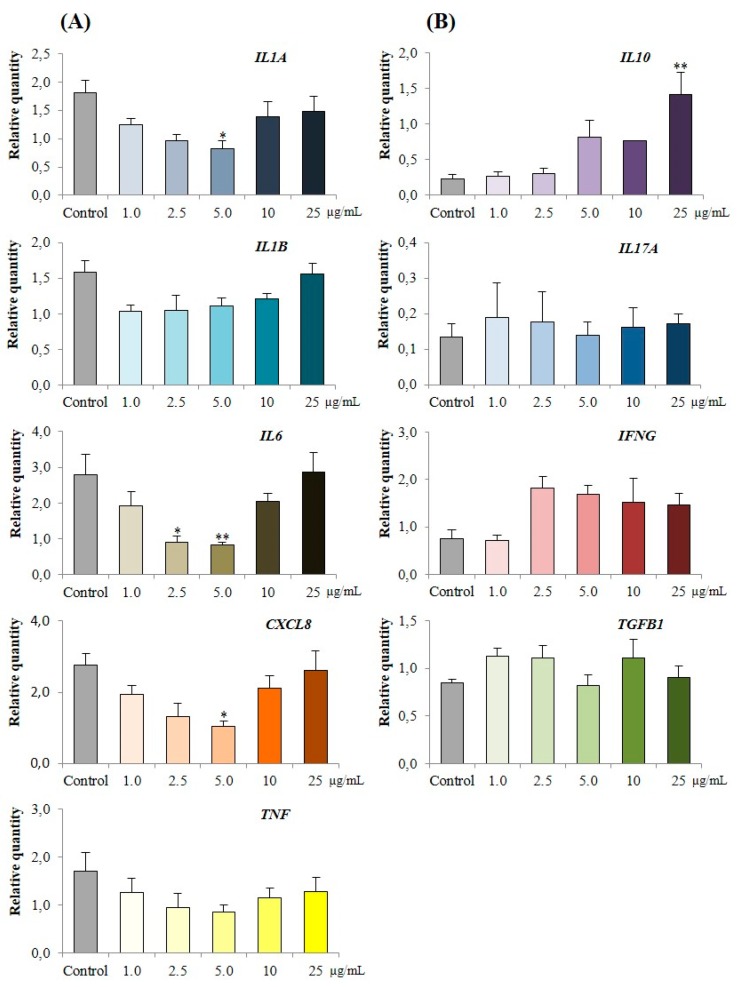
Effect of PPE co-treatment on relative quantity (RQ) of a selected panel of porcine cytokine genes in porcine colonic tissue explants stimulated by LPS, 3 h post-incubation. (**A**) Positively correlated genes. (**B**) Non correlated genes. In both panels, data point “Control” represents porcine colonic tissues stimulated with LPS. After challenge with LPS and co-treatment with PPE, total RNA was extracted, cDNA synthesised and QPCR performed to quantify the relative expression of the selected panel of genes. RQ values were compared to LPS challenged control. Each data point indicates mean ± standard error of 6 independent experiments. * *p* < 0.05 and ** *p* < 0.01 in comparison to control, *n* = 6.

**Figure 6 nutrients-11-00548-f006:**
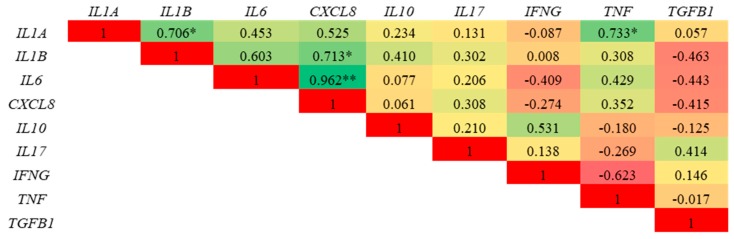
Correlation analysis of a selected panel of cytokine genes in porcine colonic tissue explants stimulated by LPS, 3 h post-incubation. Each value corresponds to the correlation coefficient of the corresponding genes. * *p* < 0.05 and ** *p* < 0.01.

**Table 1 nutrients-11-00548-t001:** Oligonucleotide sequences of forward and reverse primers used in QPCR.

	Accession Number	Forward Primer (5′–3′)	Tm (°C)	Reverse Primer (5′–3′)	Tm (°C)	Product Length (bp)	Efficiency (%)
*Reference gene*							
*ACTB*	XM_001928093.1	GCACGGCATCATCACCAA	52.75	CCGGAGCTCGTTGTAGAAGGT	55.99	70	95.02
*B2M*	NM_213978.1	CGGAAAGCCAAATTACCTGAAC	62.10	TCTCCCCGTTTTTCAGCAAA	62.20	83	103.8
*Cytokine gene*							
*IL1A*	NM_214029.1	CAGCCAACGGGAAGATTCTG	63.0	ATGGCTTCCAGGTCGTCAT	60.49	76	106.6
*IL1B*	NM_001005149.1	TTGAATTCGAGTCTGCCCTGT	60.59	CCCAGGAAGACGGGCTTT	60.94	76	104
*IL6*	AB194100	AGACAAAGCCACCACCCCTAA	55.27	CTCGTTCTGTGACTGCAGCTTATC	59.92	69	99.99
*CXCL8*	NM_213867.1	TGCACTTACTCTTGCCAGAACTG	61.9	CAAACTGGCTGTTGCCTTCTT	61.7	82	95.7
*IL10*	NM_214041.1	GCCTTCGGCCCAGTGAA	63.4	AGAGACCCGGTCAGCAACAA	63.1	71	95.7
*IL17A*	NM_001005729.1	CCCTGTCACTGCTGCTTCTG	60.57	TCATGATTCCCGCCTTCAC	60.40	57	101.2
*IFNG*	NM_213948.1	TCTAACCTAAGAAAGCGGAAGAGAA	61.12	TTGCAGGCAGGATGACAATTA	61.54	81	94.4
*TNF*	NM_214022.1	TGGCCCCTTGAGCATCA	62.5	CGGGCTTATCTGAGGTTTGAGA	62.8	68	91.5
*TGFB1*	NM_214015.1	AGGGCTACCATGCCAATTTCT	60.63	CGGGTTGTGCTGGTTGTACA	61.68	101	93

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
