# Peer review of "Anti-Inflammatory Effects of Pomegranate Peel Extracts on In Vitro Human Intestinal Caco-2 Cells and Ex Vivo Porcine Colonic Tissue Explants"

_nutrients, 2019, doi:10.3390/nu11030548_

Round 1
Reviewer 1 Report
The goal of this study was the investigation of the anti-inflammatory activity of a standardized pomegranate peel extract on two different system models. The authors of the manuscript ID 451356 performed a good work and investigation by using adequate experimental methodologies. This manuscript reports findings of potential interest that are not been built before.
Specific Comments
Major point:
A) The study is well designed; however, the authors should briefly clarify the rationalefor having chosen the concentrationsfrom 1 to 25 ug/mL. Are the concentrations physiologically reachable by supplements in humans?
B) Has been the viability of the TNF-alpha stimulated cells evaluated? Can results be affected by a different cell growth due to the pro-inflammatory stimulus?
Minor points:
A) I suggest changing the numbers’ paragraphs in the Materials and Methods section as follows:
2.2. Caco-2 ..
2.2.1. Cytotoxicity ..
2.2.2. In vitro ..
2.3. Ex vivo porcine ..
2.3.1. Ex vivo interleukin ..
2.4. RNA extraction ..
2.5. Quantitative ..
2.6 Statistical ..
B) Figure 3 and Figure 5: Please use the same symbols for statistical significance.
C) Please check the use of acronyms trough the manuscript.
D) Please carefully check the Author guidelines for the manuscript preparation to be published in the Nutrients.
Author Response
Reviewer 1
Comments and Suggestions for Authors
The goal of this study was the investigation of the anti-inflammatory activity of a standardized pomegranate peel extract on two different system models. The authors of the manuscript ID 451356 performed a good work and investigation by using adequate experimental methodologies. This manuscript reports findings of potential interest that are not been built before.
Specific Comments
Major point:
A1) The study is well designed…
We are grateful to the Reviewer who appreciated the experimental design, our original results and their potential interest for future studies.
.….. however, the authors should briefly clarify the rationale for having chosen the concentrations from 1 to 25 mg/mL.
We have chosen the range of concentration from 1.0 to 25 mg/mL of PPE on the basis of literature data reporting that in Caco-2 cell lines, concentrations higher than 25 mg/mL were cytotoxic (Omar U, Aloqbi A, Yousr M, Howell N. Effect of punicalagin on human colon cancer Caco-2 cells. Mal. J. Nutr. 2016, 22, 125-136). In addition, we observed the same results both in bovine cells BME UV1 (Mastrogiovanni, F.; Bernini, R.; Basiricò, L.; Bernabucci, U.; Campo, M.; Romani, A.; Santi, L.; Lacetera, N. Antioxidant and anti-inflammatory effects of pomegranate peel extracts on bovine mammary epithelial cells BME-UV1. Nat. Prod. Res. 2018. doi:10.1080/14786419.2018.1508149) and bovine peripheral blood mononuclear cells PBMCs (Paper in preparation).
A2) Are the concentrations physiologically reachable by supplements in humans?
The question is very interesting but the quantification of the concentrations physiologically reachable by food supplements requires specific studies on the absorption and metabolism of each bioactive compounds present into PPE. These aspects could be the object of further investigations; however, the current research represents an initial study to screen and identify the optimal concentration of PPE with anti-inflammatory activity in in vitro and ex vivo models.
B) Has been the viability of the TNF-alpha stimulated cells evaluated? Can results be affected by a different cell growth due to the pro-inflammatory stimulus?
The cytotoxicity of TNFa on Caco-2 cells was reported by Mehran and coworkers (Mehran, M.; Seidman, E.; Marchand, R.; Gurbindo, C.; Levy, E. Tumor necrosis factor-alpha inhibits lipid and lipoprotein transport by Caco-2 cells. Am J Physiol. 1995, 269, G953-G960.doi: 10.1152/ajpgi.1995.269.6.G953). This paper has clearly shown that 500 ng/mL of TNFa does not induce any cytotoxicity or anti-proliferative effect on Caco-2 cells. Based on these results, in our laboratories we currently use 10 nmol of TNFa on Caco-2 cells (see: Bahar, B.; O’Doherty, J.V.; Maher, S.; McMorrow, J.; Sweeney, T. Chitooligosaccharide elicits acute inflammatory cytokine response through AP-1 pathway in human intestinal epithelial-like (Caco-2) cells. Mol. Immunol. 2012, 51, 283-291.doi:10.1016/j.molimm.2012.03.027; Mukhopadhya, A.; Noronha, N.; Bahar, B.; Ryan, M.T.; Murray, B.A.; Kelly, P.M.; O’Loughlin, I.B.; O’Doherty, J.V.; Sweeney, T. Anti‐inflammatory effects of a casein hydrolysate and its peptide‐enriched fractions on TNFα-challenged Caco-2 cells and LPS-challenged porcine colonic explants. Food Sci. Nutr. 2014, 2, 712-723.doi: 10.1002/fsn3.153).
Minor points:
A) I suggest changing the numbers’ paragraphs in the Materials and Methods section as follows:
2.2. Caco-2 ..
2.2.1. Cytotoxicity ..
2.2.2. In vitro ..
2.3. Ex vivo porcine ..
2.3.1. Ex vivo interleukin ..
2.4. RNA extraction ..
2.5. Quantitative ..
2.6 Statistical ..
As requested by the Reviewer, we have changed the numbers’ paragraph in the “Materials and Methods” section. Similarly, we have modified the numbers’ paragraphs in the Results section.
B) Figure 3 and Figure 5: Please use the same symbols for statistical significance.
According to the request of the Reviewer, we have checked all symbols in Figures 3 and 5.
C) Please check the use of acronyms trough the manuscript.
As requested by the Reviewer, we have checked the acronyms included in the manuscript. Please note that we have used the acronyms IL1A, IL1B, IL6, IL8, IL10, IL17A, IFNG, TNF, TGFB, B2M, ACTB (Italic style) for the gene expression while IL8, TNFa (Normal style) for proteins as in our previous papers.
D) Please carefully check the Author guidelines for the manuscript preparation to be published in the Nutrients.
As requested by the Reviewer, we have carefully checked the Author guidelines of Nutrients journal.

Reviewer 2 Report
Researchers of this article tried to demonstrate the anti-inflammatory effect of pomegranate peel extracts on human colonic line and porcine colonic tissue explants. They showed the peel extract decreased the expression levels of LPS induced interleukins (IL-8) and TNF-a CACO2 and porcine colonic tissue explants. Using these observations, they proposed a possible potential role in IBD.
Minor corrections:
1. Try to change the title of research paper. Include potential role in preventing IBD.
2. Please cite these 2 manuscripts in first paragraph about polyphenols role in preventing cancer. (DOI: 10.1158/1541-7786.MCR-15-0360 Published March2016; https://doi.org/10.3892/ijo.2017.4167 Published October 19, 2017). Elaborate the first sentence to prevention of particular disease and readjust reference to each particular disease.
3. Results section 3.1, elaborate this section. Find any research available on therapeutic effects of these compounds.
4. Please discuss this research article about inflammation, IBD and curcumin (natural compound) in discussion section. This will give more strength to this paper. (https://doi.org/10.1016/j.jconrel.2018.08.004)
Author Response
Reviewer 2
Comments and Suggestions for Authors
Researchers of this article tried to demonstrate the anti-inflammatory effect of pomegranate peel extracts on human colonic line and porcine colonic tissue explants. They showed the peel extract decreased the expression levels of LPS induced interleukins (IL-8) and TNF-a CACO2 and porcine colonic tissue explants. Using these observations, they proposed a possible potential role in IBD.
Minor corrections:
1. Try to change the title of research paper. Include potential role in preventing IBD.
We thank the Reviewer for this suggestion; however, we prefer not to include the reference to IBD in the title, being our research preliminary in this field. However, we propose to modify a part of the title to better specify the content of our in vitro and ex vivo studies as follow: “Anti-inflammatory effects of pomegranate peel extracts on in vitro human intestinal Caco-2 cells and ex vivo porcine colonic tissue explants” instead of “Anti-inflammatory effects of pomegranate peel extracts on a human colonic cell line and porcine colonic tissue explants”.
To better explain that our study is preliminary, we also modified the last sentence of the Conclusion section as follow: “These data highlighted the anti-inflammatory properties of PPE in cells and tissues of the gastrointestinal tract and represent preliminary results for future studies focused on the prevention and treatment of IBD with pomegranate extracts rich in punicalagins”.
2a. Please cite these 2 manuscripts in first paragraph about polyphenols role in preventing cancer. (DOI: 10.1158/1541-7786.MCR-15 0360 Published March2016; https://doi.org/10.3892/ijo.2017.4167 Published October 19, 2017).
As requested by the Reviewer, we have cited the two selected manuscripts in the Introduction section (see Ref. 9 and 10). See also point 2b.
- Dachineni, R.; Ai, G.; Kumar, D. R.; Sadhu, S. S.; Tummala, H.; Bhat, G.J. Cyclin A2 and CDK2 as novel targets of aspirin and salicylic acid: a potential role in cancer prevention. Mol. Cancer Res. 2016, 14, 241-252. doi: 10.1158/1541-7786.
- Dachineni, R.; Kumar, D. R.; Callegari, E.; Kesharwani, S. S.; Sankaranaray, A.; Seefeldt, T.; Tummala, H.; Bhat, G.J. Salicylic acid metabolites and derivatives inhibit CDK activity: Novel insights into aspirin's chemopreventive effects against colorectal cancer. Int. J. Oncol. 2017, 51, 1661-1673. doi: 10.3892/ijo.2017.4167.
2b. Elaborate the first sentence to prevention of particular disease and readjust reference to each particular disease.
As requested by the Reviewer, we have modified the first part of the Introduction as follow: “Many epidemiological studies have demonstrated an inverse relationship between the incidence of serious diseases and consumption of fruits and vegetables containing valuable bioactive compounds according to the paradigm of the Mediterranean diet [1-4]. In particular, polyphenols have attracted great interest for their various beneficial effects on human health as the antioxidant [5, 6], anticancer activities [7-10], cardioprotective and anti-inflammatory activities [11-13] preventing the onset of cancer, chronic inflammations, and cardiovascular diseases”.
See references 1-13:
- Salas-Salvado, J.; Becerra-Tomas, N.; Garcia-Gavilan, J.F.; Bullo, M.; Barrubes, L. Mediterranean diet and cardiovascular disease prevention: what do we know? Prog. Cardiovasc. Dis. 2018, 61, 62-67. doi: 10.1016/j.pcad.2018.05.003.
- Zhao, C.-N.; Meng, X.; Li, Y.; Li, S.; Liu, Q.; Tang, G.-Y.; Li, H.-B. Fruits for prevention and treatment of cardiovascular diseases. Nutrients 2017, 9, 598; doi: 10.3390/nu9060598.
- Li, Y.; Li, S.; Meng, X.; Gan, R.Y.; Zhang, J.J; Li,, H.B. Dietary natural products for prevention and treatment of breast cancer. Nutrients 2017, 9, 728; doi:10.3390/nu9070728.
- Sergio Davinelli, S.; Maes, M.; Corbi, G.; Zarrelli, A.; Craig Willcox, D.; Scapagnini, G. Dietary phytochemicals and neuro-inflammaging: from mechanistic insights to translational challenges. Immun. Ageing 2016, 13, 16. doi:10.1186/s12979-016-0070-3.
- Ferrazzano, G.M.; Amato, I.; Ingenito, A.; Zarrelli, A.; Pinto, G.; Pollio, A. Plant polyphenols and their anti-cariogenic properties: a review. Molecules 2011, 16, 1486-1507. doi:10.3390/molecules16021486. Barontini, M.; Bernini, R.; Carastro, R.; Gentili, P.; Romani, A. Synthesis and DPPH radical scavenging activity of novel compounds obtained from tyrosol and cinnamic acid derivatives. New J. Chem. 2014, 38, 809-816. doi: 10.1039/c3nj01180A.
- Bernini, R.; Barontini, M.; Cis, V.; Carastro, I.; Tofani, D.; Chiodo, R. A.; Lupattelli, P.; Incerpi, S. Synthesis and evaluation of the antioxidant activity of lipophilic phenethyl trifluoroacetate esters by in vitro ABTS, DPPH and in cell-culture DCF assays. Molecules 2018, 23, 208. doi: 10.3390/molecules23010208.
- Bernini, R.; Merendino, N.; A. Romani, F. Velotti. Naturally occurring hydroxytyrosol: synthesis and anticancer potential. Curr. Med. Chem. 2013, 20, 655-670; doi:102174/092986713804999367.
- Bernini, R.; Gilardini Montani, M. S.; Merendino, N.; Romani, A. Velotti, F. Hydroxytyrosol-derived compounds: a basis for the creation of new pharmacological agents for cancer prevention and therapy. J. Med. Chem. 2015, 58, 9089-9107. doi:10.1021/acs.jmedchem.5b00669.
- Dachineni, R.; Ai, G.; Kumar, D. R.; Sadhu, S. S.; Tummala, H.; Bhat, G.J. Cyclin A2 and CDK2 as novel targets of aspirin and salicylic acid: a potential role in cancer prevention. Mol. Cancer Res. 2016, 14, 241-252. doi: 10.1158/1541-7786.
- Dachineni, R.; Kumar, D. R.; Callegari, E.; Kesharwani, S. S.; Sankaranaray, A.; Seefeldt, T.; Tummala, H.; Bhat, G.J. Salicylic acid metabolites and derivatives inhibit CDK activity: Novel insights into aspirin's chemopreventive effects against colorectal cancer. Int. J. Oncol. 2017, 51, 1661-1673. doi: 10.3892/ijo.2017.4167.
- Khurana, S.; Venkataraman, K.; Amanda Hollingsworth, A.; Piche, M.; Tai, T. C. Polyphenols: benefits to the cardiovascular system in health and in aging. Nutrients 2013, 5, 3779-3827. doi:10.3390/nu5103779.
- Serino, A.; Salazar, G. Protective role of polyphenols against vascular inflammation, aging and cardiovascular disease. Nutrients 2019, 11, 53. doi.org/10.3390/nu11010053.
- Yahfoufi, N.; Alsadi, N.; Jambi, M.; Matar, C. The immunomodulatory and anti-inflammatory role of polyphenols. Nutrients 2018, 10, 1618. doi:10.3390/nu10111618.
3. Results section 3.1, elaborate this section. Find any research available on therapeutic effects of these compounds.
As requested by the Reviewer, we elaborated the section 3.1 adding some literature data on the therapeutic effects of pomegranate bioactive constituents: “PPE was prepared in our laboratories from a waste of pomegranate juice production and a sample was characterized by HPLC/DAD/ESI-MS and NMR analysis. As showed in Figure 1, high-molecular weight phenols as a- and b-punicalagins were the main polyphenols with 146.9±1.465 mg/g and 266.3±1.687 mg/g, respectively, while gallic acid, ellagic acid and granatin B were the minor components. Further details about HPLC profile and 1H NMR spectrum of PPE are available in our recent papers [42,43]. The molecules found into PEE showed numerous therapeutic properties as demonstrated by in vitro, animal, and clinical trials [45]. In particular, a phase II clinical trial performed with 46 men with prostate cancer evidenced a significant decrease in serum prostate specific antigen (PSA) levels (average=27%) during treatment with pomegranate juice [45].
See the reference 45:
- Jerenka, J. Therapeutic applications of pomegranate (Punica granatum L.): a review. Altern. Med. Rev. 2008, 13, 128-144.
4. Please discuss this research article about inflammation, IBD and curcumin (natural compound) in discussion section. This will give more strength to this paper. (https://doi.org/10.1016/j.jconrel.2018.08.004).
We included the paper suggested by the Reviewer in our manuscript and the Discussion we added the following sentences: “Recent studies have investigated the possible role of diet and food supplements such as vitamins, polyphenols and micronutrients in the management of IBD [35,36]. In clinical trials, curcumin and beverages rich in polyphenols as green tea have effectively reduced the symptoms in IBD patients [35]. To the best of our knowledge, punicalagins have not been included so far in these studies even if they are recognized as the most therapeutically beneficial pomegranate constituents [45]. Based on these literature data, we projected to evaluate the anti-inflammatory effects of PPE, rich in punicalagins, on TNFα-challenged Caco-2 cells and LPS-challenged porcine colonic explants as preliminary investigations for future studies focused on the prevention and treatment of the chronic inflammation of the gastrointestinal tract and IBD.”
See references 35, 36 and 45:
- Rossi, R.E.; Whyand, T.; Murray, C.D.; Hamilton, M.I.; Conte, D.; Caplin, M.E. The role of dietary supplements in inflammatory bowel disease: a systematic review. Eur. J. Gastroenterol. Hepatol. 2016, 28, 1357-1364.
- Kesharwania, S.S.; Ahmadb, R.; Bakkaria, M.A.; Rajputa, M.K.S.; Dachineni, R.; Valivetia, C.K.; Kapurb, S.; Bhata, G.J.; Singh, A.B.; Tummala, H. Site-directed non-covalent polymer-drug complexes for inflammatory bowel disease (IBD): Formulation development, characterization and pharmacological evaluation. J. Control Release 2018, 290, 165-179.doi: 10.1016/j.jconrel.2018.08.004.
- Jerenka, J. Therapeutic applications of pomegranate (Punica granatum L.): a review. Altern. Med. Rev. 2008, 13, 128-144.
We sincerely thank you for your time and consideration,
Prof. Roberta Bernini
